# Exploring communication between parents and clinical teams following children's heart surgery: a survey in the UK

Christina Pagel,[1,2] Catherine Bull,[2] Martin Utley,[1] Jo Wray,[3] David J Barron,[4] Serban Stoica,[5] Shane M Tibby,[6] Victor Tsang,[3] Katherine L Brown[3]

For numbered affiliations see end of article.

**Correspondence to**
Professor Christina Pagel; c.pagel@ucl.ac.uk

## ABSTRACT

**Objective** To explore communication between clinicians and families of children undergoing heart surgery.

**Design** This study was part of a larger study to select, define and measure the incidence of postoperative complications in children undergoing heart surgery. Parents of children recruited to a substudy between October 2015 and December 2017 were asked to complete a questionnaire about communication during their child's inpatient stay. We explored all responses and then disaggregated by the following patient characteristics: presence of a complication, length of stay, hospital site, ethnicity and child's age. This was a descriptive study only.

**Setting** Four UK specialist hospitals.

**Results** We recruited 585 children to the substudy with 385 responses (response rate 66%). 81% of parents reported that new members of staff always introduced themselves (18% sometimes, 1% no). Almost all parents said they were encouraged to be involved in decision-making, but often only to some extent (59% 'yes, definitely'; 37% 'to some extent'). Almost two-thirds of parents said they were told different things by different people which left them feeling confused (10% 'a lot'; 53% 'sometimes'). Two-thirds (66%) reported that staff were definitely aware of their child's medical history (31% 'to some extent'). 90% said the operation was definitely explained to them (9% 'to some extent') and 79% that they were definitely told what to do if they were worried after discharge (17% 'to some extent'). Parents of children with a complication tended to give less positive responses for involvement in decision-making, consistent communication and staff awareness of their child's medical history. Parents whose children had longer stays in hospital tended to report lower levels of consistent communication and involvement in decision-making.

**Conclusions** Our results emphasise the need for consistent communication with families, particularly where complications arise or for children who have longer stays in the hospital.

## INTRODUCTION

Many children with congenital heart disease undergo heart surgery with stays on intensive care that can be prolonged, complex and very stressful for parents.[1–3] Consistent and

### What is already known on this topic?

► Paediatric cardiac surgery is a very stressful time for patients and families.
► Communication between clinicians and families and involvement in decision-making can alleviate some of the stress experienced by families.

### What this study hopes to add?

► It was common for parents of children undergoing heart surgery to report being told different things by different people (53% said it happened sometimes and 10% said a lot).
► Parents of children who stayed longer in hospital and/or experienced a complication tended to give less positive responses on consistency of communication and on being involved in decisions about their child's care.
► Clinical teams caring for children with complex conditions in tertiary settings need to focus on providing consistent information to families and involving them in decision-making, particularly for longer staying children.

effective communication with caregivers[4] and involvement in decision-making[5] have been recognised as important ways to reduce stress. However, the environment of intensive care, clinician working patterns and involvement of many professionals from different disciplines in the care of the child and family can make consistent and coherent communication challenging.

A number of aspects of communication have been associated with improved outcomes—for example, family understanding is improved by clear, jargon-free communication by clinicians and checking family understanding[6]; trust in clinicians is enhanced by personalising communication[7]; and family satisfaction with conversations with

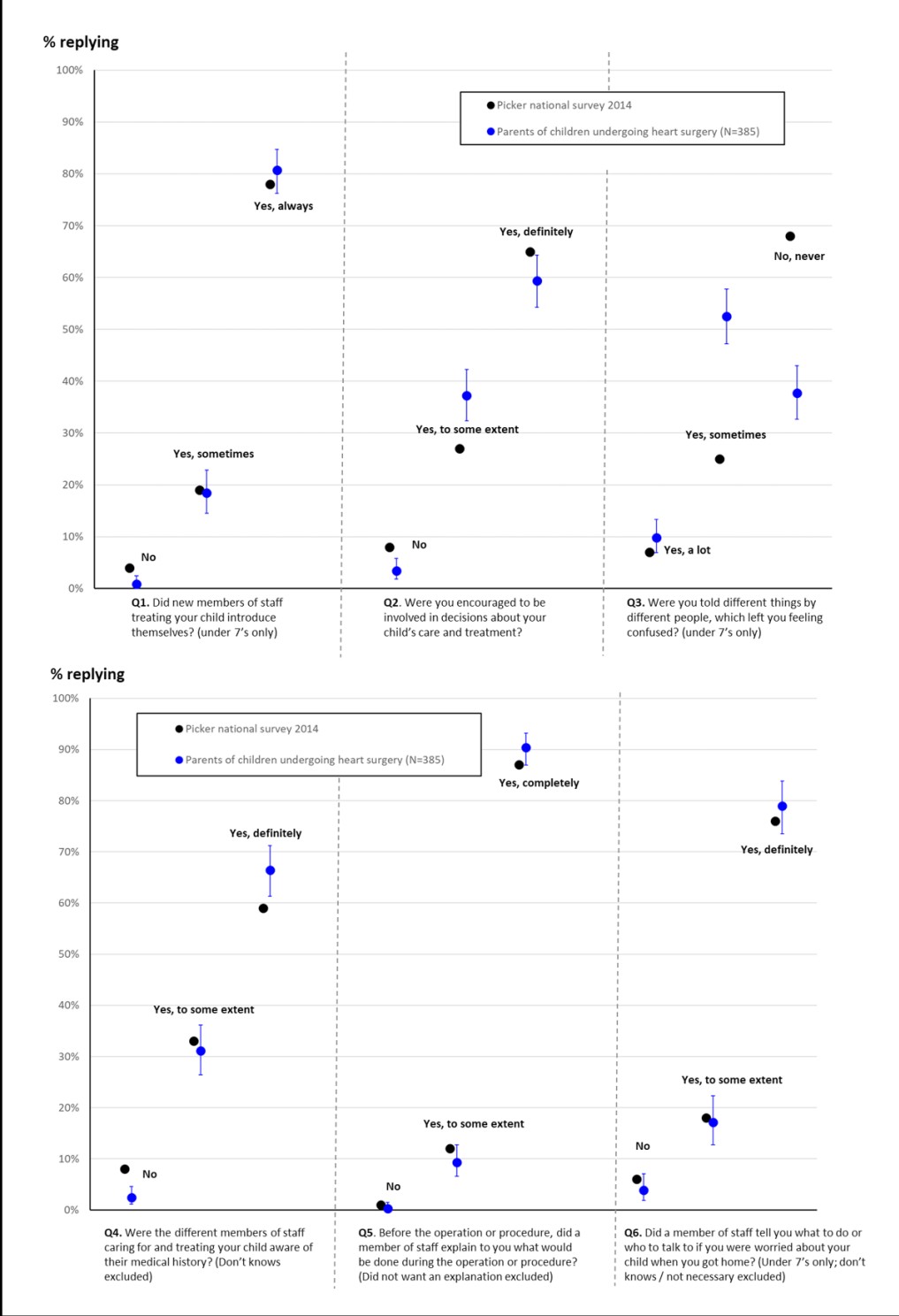

**Figure 1** Proportion of responses to each of the six questions for the national Picker survey (black dots) and our study of 385 participants, who were all parents of children who had heart surgery (blue dots). Exact 95% CI are shown for the proportions measured in our study.

health professionals is increased when family members are provided with more opportunities to talk themselves.[8] Parents' own beliefs, hopes and emotions are likely to influence their approach to decision-making for their infant in the intensive care unit although parents will often not volunteer such information unless encouraged to do so by clinicians.[9] Recent evidence suggests that

clinicians talk more than parents in family meetings, that neither clinicians nor parents address many questions to each other and that the delivery of medical information is (not surprisingly) prioritised.[9 10] However, failing to ask parents questions and determine parental understanding and perceptions may indicate a failure to identify 'information-overload', resulting in the recognised pattern of

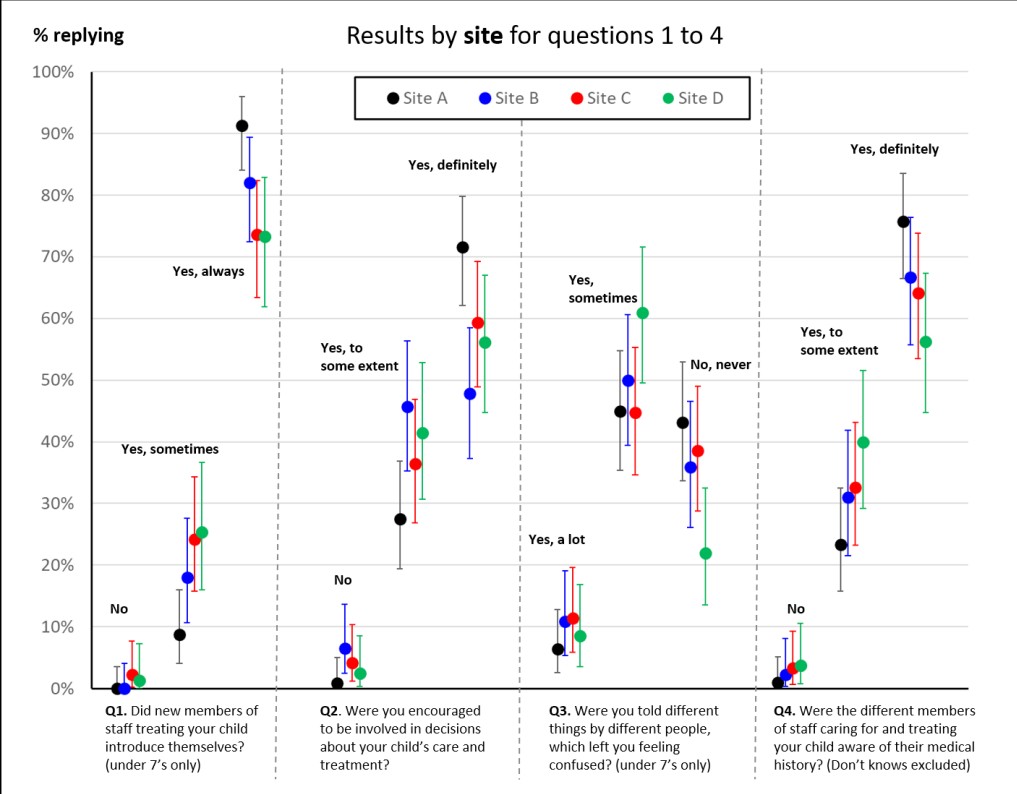

**Figure 2** Proportion of responses to questions 1–4 disaggregated by study site. We only show proportions for questions where there was a noticeable, potentially interesting, difference in responses.

variability in clinicians' and parents' recall of a shared conversation.[11] A further important consideration, particularly in situations of prolonged intensive care or hospital stay, is the finding that conversations between parents and clinicians diminish as a child's condition stabilises, which can result in feelings of loneliness and abandonment.[12] Parents also need to be provided with information about when and whom to call for advice and support once discharged from hospital to home, particularly in situations where an infant has complex congenital heart disease.[13]

From January 2014 to June 2018, we carried out a large multicentre UK study to identify, categorise and then measure the incidence and impact in the hospital and beyond discharge of important early complications among paediatric cardiac surgery patients (NIHR HS&DR 12/5005/06[14]). As part of this study, we documented parents' views about their communication with their child's clinical team at each participating specialist centre. Our aim was to explore how parents of children undergoing heart surgery perceived communication with their clinical team and the potential association of patient characteristics with the reported quality of communication.

## METHODS

We have broken the methods section into subsections that cover:

► The larger study, including how the complications studied were selected for measurement and defined.
► How the communication survey was administered and to whom.
► How we analysed the results.
► Ethical approvals and patient involvement.

### Larger study overview to select, define and measure complications following children's heart surgery

The larger study included prospective monitoring of consecutive cardiac surgery cases at 5 of the UK's 10 paediatric cardiac specialist centres over 21 months, from October 2015 to June 2017. A subset of children in each centre was recruited to an impact substudy, where parents consented to a 6-month follow-up including assessment of the quality of life and clinical outcomes, to compare outcomes between children who did and did not experience a complication.

In the overall study, we tracked children undergoing 3090 consecutive procedures for the following nine complications: acute neurological event; extracorporeal life support; feeding problems; major adverse events; nectrotising enterocolitis; postsurgical infection; prolonged pleural effusion; renal replacement therapy; and unplanned re-intervention. A full write-up of the selection[15] and definition[16] processes of all measured complications and communication are available in the online supplementary files.

A key aim of the larger study was to incorporate a broad set of perspectives, including those from family representatives and professionals from different sectors on what early complications were important to monitor in routine practice. We convened a study selection panel which met twice in 2014 to select up to 10 complications for prospective monitoring. At the second selection panel, 10 complications were chosen, including 'poor communication between the clinical team and family'. There was some resistance from clinicians on the panel to the inclusion of poor communication as something that could not be fairly attributed to the surgical act; however, family representatives and others successfully argued that poor communication: has a potentially significant impact on the family and their ability to care for their child once home; is associated with having surgery; and is also potentially mitigated through improved practice.

We convened a separate definitions panel which met twice in 2014, following the first and second meetings of the selection panel, respectively. The panel included three paediatric cardiac surgeons, three congenital cardiologists, three paediatric intensive care specialists and two specialist nurses. The definitions panel aimed to establish the diagnostic criteria that constitute the definition of each of the chosen complications and then define the measurement protocol for each of the complications. The definition panel considered that there was the potential to define poor communication between the treating team and family in the future, but that it would necessarily involve asking parents about their experience in a way that would involve new data collection, which was not feasible for the part of the study which followed all children using routine data collection. This reduced the number of complications tracked for all children to nine. Instead, the definitions panel recommended that communication should be assessed only in the subset of patients whose parents had been formally consented

into the impact study, as part of the postoperative study follow-up.

In 2014, the Picker Institute Europe, funded by the UK Care Quality Commission, carried out England's first national survey of family experiences of paediatric stays in the hospital. Questionnaires were sent to families of all children discharged from all hospitals in England during August 2014, with over 19 000 responses (response rate 27%).[17] The Picker Survey included several questions around communication and provides national baseline metrics for standards of communication between staff, parents and patients for hospital inpatient stays. The definitions panel worked with the Picker Institute to identify six questions from their national survey to ask study parents about communication that captured a range of aspects of communication between families and clinical teams. The panel decided not to set a threshold for what defines 'poor communication'; instead, they recommended exploration of the responses among control and case patients in the impact study and possible associations with other clinical factors as part of secondary data analysis.

The selected questions and comments are shown in table 1.

### Population and administering the survey

The wider cardiac study prospectively monitored all children under 16 years of age who had cardiac surgery within five participating tertiary care hospitals between 1 October 2015 and 30 June 2017 and tracked the incidence of the nine selected postoperative complications. Families of children with these complications and controls without a complication were then approached for consent to participate in a 6-month follow-up programme. Consenting families were given the communication questionnaire either at or shortly after discharge, along with a stamped addressed return envelope. Parents

**Table 1** Questions chosen for the communication survey. The authors noted that 21/385 children were over 7 years of age in the study (5.5%)

| Question | Possible responses | Notes |
|---|---|---|
| Q1. Did new members of staff treating your child introduce themselves? | No; yes, sometimes; yes, always | Picker restricted this to under 7 year olds only. We did the same for our analysis |
| Q2. Were you encouraged to be involved in decisions about your child's care and treatment? | No; yes, to some extent; yes, definitely | |
| Q3. Were you told different things by different people, which left you feeling confused? | Yes, a lot; yes, sometimes; no, never | Picker restricted this to under 7 year olds only. We did the same for our analysis |
| Q4. Were the different members of staff caring for and treating your child aware of their medical history? | No; yes, to some extent; yes, definitely | |
| Q5. Before the operation or procedure, did a member of staff explain to you what would be done during the operation or procedure? | No; yes, to some extent; yes, completely; do not know; not necessary | Picker excluded the 'do not know/not necessary' from their results. We did the same for our analysis. |
| Q6. Did a member of staff tell you what to do or whom to talk to if you were worried about your child when you got home? | No; yes, to some extent; yes, definitely; do not know; not necessary | Picker excluded the 'do not know/not necessary' from their results. We did the same for our analysis |

were asked complete and return it without personal identifiers within 6 weeks of their child's operation. Parents were asked on their child's first stay in the hospital during the study period only and returned no more than one questionnaire (so all questionnaires in the study relate to different children).

Parents were assured that no clinicians caring for their child or researchers would see their responses. Questionnaire responses were entered by a separate data entry clerk with access only to the child's pseudonymised study ID. The communication survey data entry screen was password protected to prevent clinician or research team's access. For staff resource reasons, only four of the five hospitals participated in the communication survey.

### Data analysis
We performed a descriptive analysis only. We first compared the proportion of each response received for each question in our impact substudy with the corresponding proportions from the 2014 National Picker Survey.

We then explored responses from parents in the impact substudy disaggregated by: occurrence of a postoperative complication; length of stay in hospital; ethnicity; age of the child and hospital site.

We calculated 95% CIs for all responses for each question in our substudy group. All data were analysed using Stata software V.14. Note that as this is a descriptive study only, we do not suggest that our results have statistical significance.

### Patient involvement
Families were involved in the broader study to select the complications being measured by: taking part in in-person focus groups in three cities to discuss the impact of complications following heart surgery; taking part in an online forum over 10 weeks run by the Children's Heart Federation that explored perspectives on complications; three family representatives were members of the formal selection panel that selected the final nine complications. This substudy on communication reported in this paper was motivated directly by reports from families

highlighting the importance and challenges of communication after children's heart surgery. The project team shared several newsletters with families of children undergoing heart surgery over the course of the study reporting on progress and outcomes. Newsletters were disseminated via the Children's Heart Federation.

## RESULTS
We recruited 585 children to the impact study at the four centres participating in the communication survey. We received 385 responses (response rate 66%). The median age at procedure was 2.6 months (IQR: 10 days–11 months) and the median length of stay was 16 days (IQR 9–26 days). The breakdown of the characteristics of children by the site is given in table 2.

### Comparison to the national, all specialty inpatient survey
Responses to each of the six questions in the national survey and our survey of parents of children who had heart surgery are shown in table 3 and figure 1. We reiterate that in describing potentially interesting differences in the results, we are not asserting statistical significance.

Parents' responses were very similar to the national survey for questions 1, 5 and 6. However, parents in our study reported slightly lower rates of involvement in decision-making (Q2); much lower rates of consistent communication (Q3) and slightly higher rates of awareness of their child's medical history (Q4). In particular, for consistent communication, only 38% of parents in our study said that they were never told different things by different people compared with 68% of parents in the national survey.

### Disaggregation by different patient characteristics
Responses to our survey broken down by site; child's age; length of stay; and presence of a complication are shown in figures 2–5 for each characteristic where there was an observed, potentially interesting, difference in responses. Note that we do not show results by ethnicity because results were broadly similar across the questions and ethnic status.

**Table 2** Characteristics of children recruited for the communication study at each site and for non-respondents

| Characteristic | Site A | Site B | Site C | Site D | Overall | All non-respondents |
|---|---|---|---|---|---|---|
| Total responses | 115 | 92 | 96 | 82 | 385 | 200 |
| Black or minority ethnic (%) | 23 | 35 | 23 | 7 | 22 | 24 |
| Complication (%) | 50 | 61 | 46 | 49 | 51 | 52 |
| Neonate (<30 days) | 51% | 42% | 27% | 34% | 39% | 37% |
| Infant | 32% | 43% | 36% | 33% | 36% | 39% |
| Child (>1 year) | 17% | 15% | 37% | 33% | 25% | 24% |
| Length of stay a week or less | 20% | 13% | 10% | 7% | 13% | 17% |
| Length of stay 1–3 weeks | 46% | 54% | 58% | 63% | 55% | 45% |
| Length of stay over 3 weeks | 34% | 33% | 32% | 29% | 32% | 38% |
| Child over 7 (excluded from Q1 and Q3 analysis) | 5% | 3% | 5% | 9% | 5% | 6% |

**Table 3** Results of our prospective survey. Quantities are percentage of respondents, with 95% Cs for each proportion. Following the national survey, we excluded 'Don't know/unnecessary' responses from Q5 and Q6. This comprised 0.5% and 7% of responses respectively

| | Q1. Did new members of staff treating your child introduce themselves? | | | Q2. Were you encouraged to be involved in decisions about your child's care and treatment? | | | Q3. Were you told different things by different people, which left you feeling confused? | | |
| --- | --- | --- | --- | --- | --- | --- | --- | --- | --- |
| | No | Yes, sometimes | Yes, always | No | Yes, to some extent | Yes, definitely | Yes, a lot | Yes, sometimes | No, never |
| 385 parents of children who have had heart surgery (% with 95% CI) | 1 (0 to 2) | 18 (15 to 23) | 81 (76 to 85) | 3 (2 to 6) | 37 (32 to 42) | 59 (54 to 64) | 10 (7 to 13) | 53 (47 to 58) | 38 (33 to 43) |

| | Q4. Were the different members of staff caring for and treating your child aware of their medical history? | | | Q5. Before the operation or procedure, did a member of staff explain to you what would be done during the operation or procedure? | | | Q6. Did a member of staff tell you what to do or whom to talk to if you were worried about your child when you got home? | | |
| --- | --- | --- | --- | --- | --- | --- | --- | --- | --- |
| | No | Yes, to some extent | Yes, definitely | No | Yes, to some extent | Yes, completely | No | Yes, to some extent | Yes, definitely |
| 385 parents of children who have had heart surgery (% with 95% CI) | 2 (1 to 5) | 31 (26 to 36) | 66 (61 to 71) | 0 (0,1) | 9 (7 to 13) | 90 (87 to 93) | 4 (2 to 7) | 17 (13 to 22) | 79 (73 to 84) |

Quantities are the percentage of respondents, with 95% CIs for each proportion. Following the national survey, we excluded 'Do not know/unnecessary' responses from Q5 and Q6. This comprised 0.5% and 7% of responses, respectively

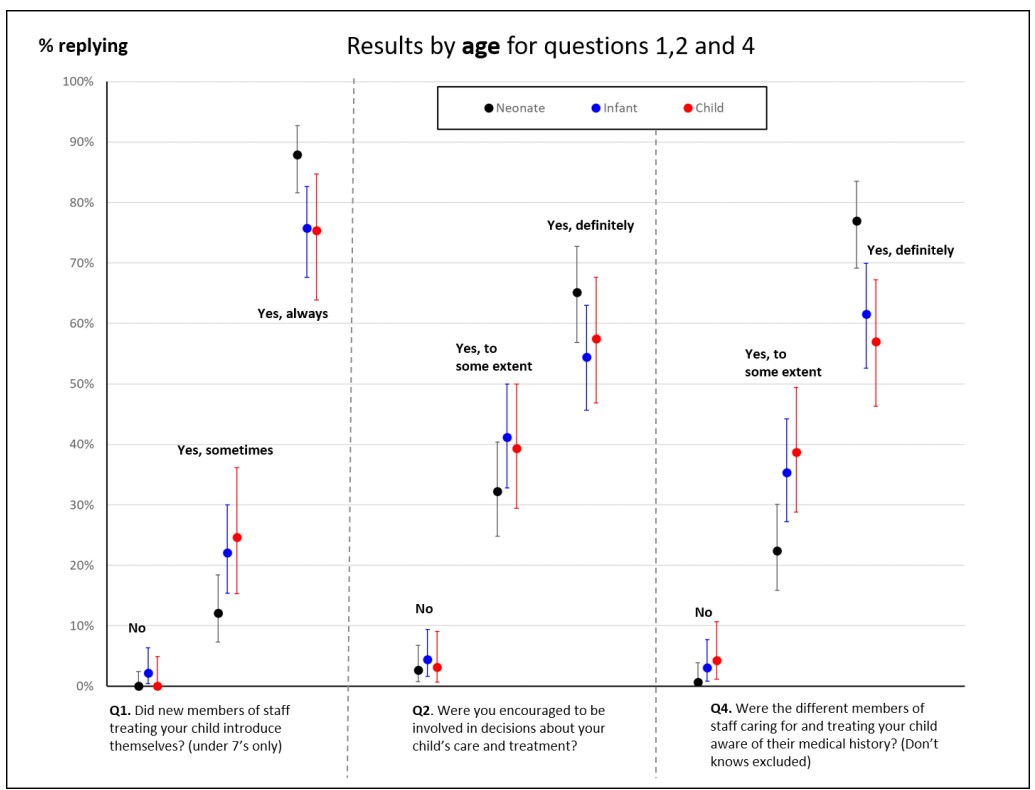

**Figure 3** Proportion of responses to questions 1, 2 and 4 disaggregated by the age of the child (neonate, infant [1 month–1 year] and child, over 1 year old). We only show proportions for questions where there was a noticeable, potentially interesting, difference in responses.

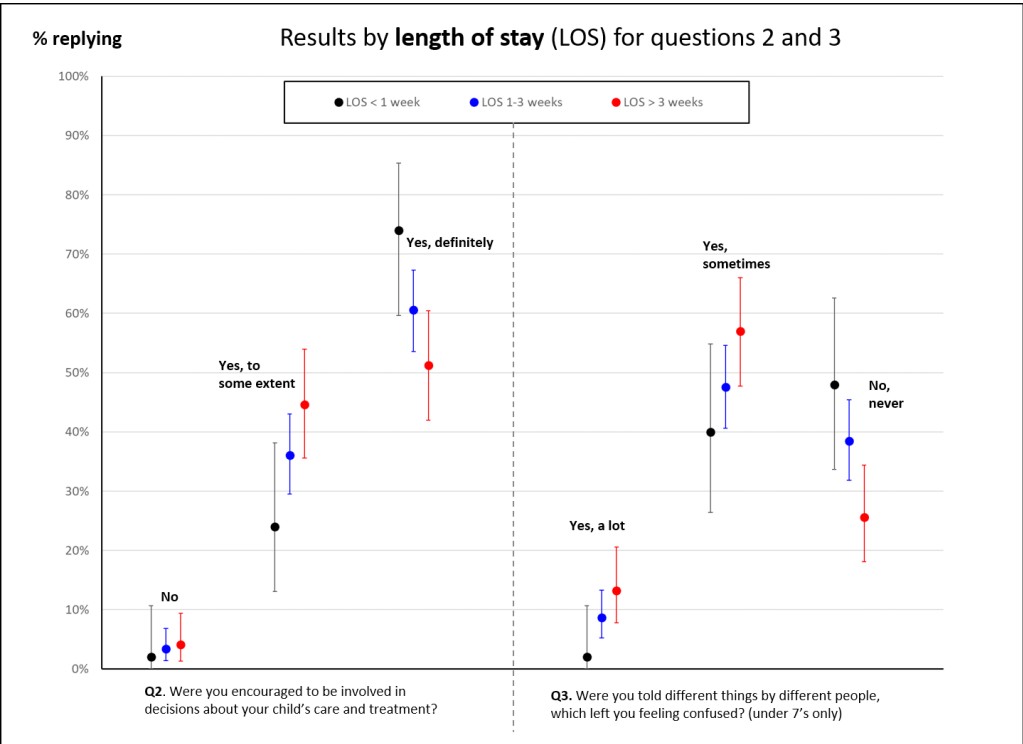

**Figure 4** Proportion of responses to questions 2 and 3 disaggregated by the length of stay. We only show proportions for questions where there was a noticeable, potentially interesting, difference in responses.

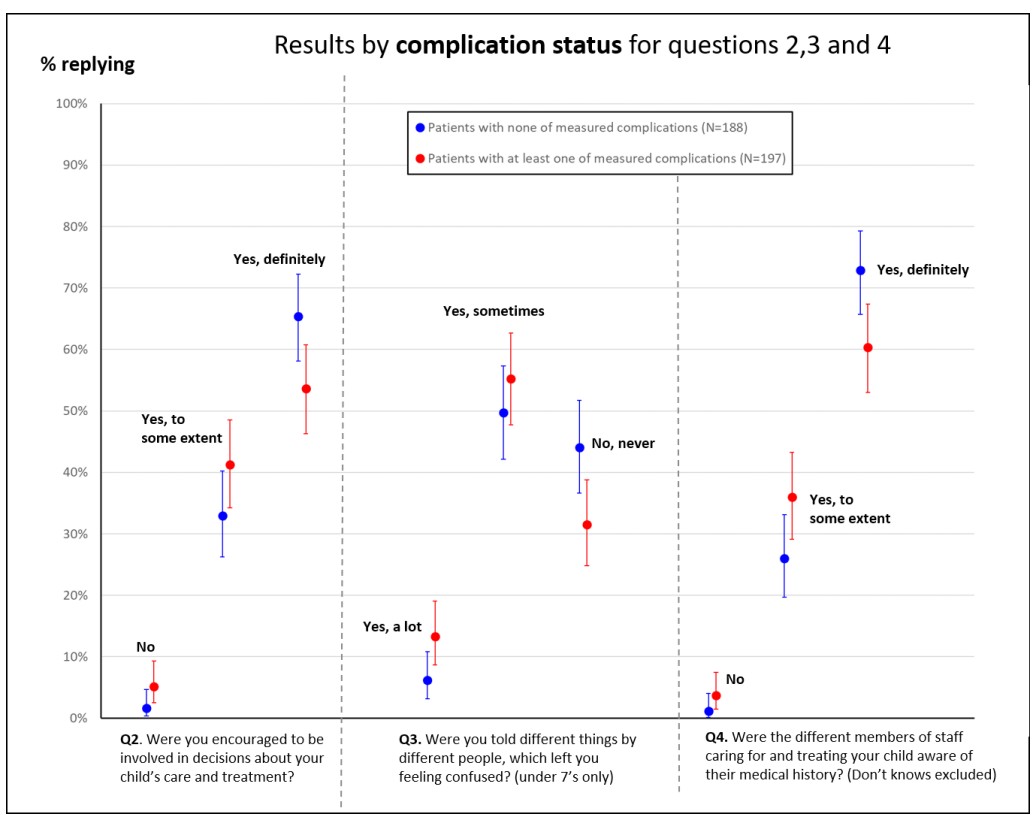

**Figure 5** Proportion of responses to questions 2–4 disaggregated by whether the child experienced a complication. We only show proportions for questions where there was a noticeable, potentially interesting, difference in responses.

Within the study group, parents from site A tended to give more positive responses than the other sites on questions 1–4 (figure 2), particularly for always introducing themselves (over 90% for site A, almost 20% higher than for sites C and D) and involvement in decision-making. Parents of neonates tended to give more positive responses for staff introducing themselves, involvement in decision-making and staff knowing their child's medical history compared with parents of infants or older children (figure 3).

Parents of children who stayed longer in hospital tended to give less positive responses to being involved in decisions around care and receiving consistent communication (figure 4). Where children had experienced a complication after surgery (figure 5), parents tended to give less positive responses for involvement in decision-making, receiving consistent communication and staff awareness of their child's medical history.

## DISCUSSION

Parental expectations of the care experience in tertiary-care paediatric services are high. Parents of children having heart surgery had very positive responses, in line with the UK national all-specialty results, for three of the six questions: staff introducing themselves, explaining the operation and letting parents know what to do once home after discharge. The question asking whether parents were told different things by different people received the least positive responses, with only 38% saying that it never happened. Possible explanations are that heart surgery is a particularly complex treatment involving a larger than average clinical care team and that a child's course of recovery can be very variable. Additionally, children undergoing heart surgery have longer stays in hospital than the overall paediatric inpatient population and parents typically spend a lot of time by their child's bedside, providing more opportunities for confusing communication to arise. However understandable it is, it nonetheless represents an important issue for clinical teams to be aware of.

Unsurprisingly, parents of neonates tended to report that staff were more aware of their child's (intrinsically shorter) medical history, but parents of children who experienced complications reported lower awareness among staff of their child's medical history. Possible explanations could be that complications constitute changes to the medical history and/or because associated parental stress means that information is harder to process.[11] Parents of neonates also tended to report more involvement in decision-making and that staff always introduced themselves—could there be a conscious or subconscious effort to make extra efforts for parents of newborns? On the other hand, parents of children who stayed longer or who experienced a complication reported lower levels of involvement in decision-making and consistent communication. Both longer stays and complications are indicators of a more difficult recovery which can encompass changing treatment decisions, which could contribute to parents being confused about what was happening and feeling disempowered in decision-making.

This study represents an initial, descriptive, exploration of different aspects of communication experienced by parents of children undergoing heart surgery. We did not prospectively define measures of 'poor communication' or test any hypotheses. The responses from the 2014 national survey were used for context but with the limitation that our population tends to be younger and much sicker than the average paediatric inpatient. Furthermore, the patient characteristics explored for associations with communication in our population are not independent. For instance, neonates are more likely to experience complications and children who experience complications, tend to stay longer in the hospital. Thus, we cannot draw definitive conclusions from our results but they do suggest areas for further study and reflection.

One potential area for further research is in enabling consistent communication with parents and understanding better the barriers to doing so in the congenital heart disease setting. Designating dedicated members of staff assigned to each family or further training teams to avoid 'information overload' could be potential positive interventions.[11] That improved communication is possible and worth further investigation of possible modifying factors is suggested by the consistently higher proportion of responses recorded at site A across questions 1–4. For instance, the structure of intensive care unit staffing, the physical layout of a unit and the ratio of the volume of admissions to staff may all affect parents' experience of communication with their child's clinical team. Our prospective survey across four sites represents an important early step to improving the quality of communication with families.

### Author affiliations

[1]Clinical Operational Research Unit, University College London, London, UK
[2]Department of Paediatric Intensive Care, Great Ormond Street Hospital NHS Foundation Trust, London, UK
[3]Heart and Lung Division, Great Ormond Street Hospital for Children NHS Foundation Trust, London, UK
[4]Department of Intensive Care and Paediatric Cardiac Surgery, Birmingham Children's Hospital NHS Foundation Trust, Birmingham, UK
[5]Department of Intensive Care and Paediatric Cardiac Surgery, Bristol Children's Hospital, Bristol, UK
[6]Department Paediatric Cardiology and Cardiac Surgery, Evelina London Children's Hospital, Guy's and St Thomas' NHS Foundation Trust, London, UK

**Contributors** KB and VT: principal investigators (PIs) for the study and responsible for data collection at one hospital. SMT, DB and SS: are local responsible for data collection and study oversight at their hospital site. CP, CB, MU, JW: designed the communication substudy. CP: carried out the analysis of the communication data and wrote the first draft of the paper. All authors apart from CB are co-applicants of the study and designed the overall programme of research. All authors read and commented on the final draft.

**Funding** This project was funded by the National Institute for Health Research Health Services and Delivery Research Programme (project number 12/5005/06). KB and JW were supported by the National Institute for Health Research Biomedical Research Centre at Great Ormond Street Hospital for Children NHS Foundation Trust and University College London.

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
