## [Reviewer comments · BMJ Paediatrics Open]

ARTICLE DETAILS

TITLE (PROVISIONAL)	Exploring communication between parents and clinical teams following children's heart surgery: a survey in the UK
AUTHORS	Pagel, Christina; Bull, Catherine; Utley, Martin; Wray, Jo; Barron, David; Stoica, Serban; Tibby, Shane; Tsang, Victor; Brown, Katherine

VERSION 1 – REVIEW

REVIEWER	Reviewer name: Peter Flom Institution and Country: Peter Flom Consulting, USA Competing interests: None
REVIEW RETURNED	14-Nov-2018

GENERAL COMMENTS	I confine my remarks to statistical aspects of this paper. Since the patients were clustered within hospitals and since patients in a particular hospital are likely to share some similarities, the normal logistic regression model may not be appropriate. Table 2 shows large differences by site, so, the authors should use a nonlinear multi-level model. I don't really like the binarization of questions. Surely most of these questions are not "always" vs. "not always" but vary along a continuum? Unless reason can be shown for binarization (e.g. that almost no patients gave any answer other than "always") this is a mistake that leads to loss of power and increased type I error. Rather than test each question separately, it might make sense to combine them using factor analysis. This would greatly increase statistical power, not only because the combined measure would have less error but because the Bonferroni correction would be /2 rather than /12. In addition, this would make the response more-or-less continuous. Taking the log of age does not really address the problem of nonlinearity. It would be better to use a spline of age.
---

REVIEWER	Reviewer name: Erica Sood, PhD Institution and Country: Nemours/Al duPont Hospital for Children, USA Competing interests: None
REVIEW RETURNED	28-Nov-2018

GENERAL COMMENTS	This study aims to compare aspects of communication between families and medical teams for pediatric cardiac surgery patients to a national, all specialty survey. While communication is a very important aspect of care in pediatric cardiology practice, several concerns about methodology and study limitations limit the interpretability and utility of findings.
--

Abstract:

1. The abstract is likely to be confusing to a reader not already familiar with the overall study, impact substudy, and the various samples. Also, multiple paragraphs in one section of the abstract is not typical. Please consider editing for clarity.
2. The second point in “what this study adds” should be clarified and expanded upon. Confusion was not necessarily measured after the full question was changed into a binary question. This statement may be a misrepresentation of findings. It may be more accurate to say that 2/3 of parents reported that they were not always told consistent information.

Introduction:

1. The literature review in the introduction provides a good rationale for importance of study. Aspects of communication identified in introduction link well with survey questions.
2. There is no justification provided for separating patients in the impact study by morbidity status. Please provide background literature to support this decision.
3. Portions of the introduction would be better suited for the methods section (e.g., background on Pickler survey).
4. Please include a statement about the knowledge gaps that this study intends to address. Why is this particular study needed? Please also include the research questions and any hypotheses in introduction.

Methods:

1. The demographic information that was included in the paper is insufficient for fully understanding the results of this study. For example, who were the participants included in the Pickler survey (e.g., inclusion of “control” participants in Pickler sample? gender, age, length of hospital stay, etc. [include table of demographic information]; were patients with CHD included in Pickler sample?). Demographic information on participants in impact study are also critical (e.g., gender, number of morbidities, etc.) as well as a formal comparison of demographic characteristics of participants across 4 impact sites and Pickler study to assess for systematic differences that could have impacted the internal validity.
2. The use of the word “morbidities” is variable throughout the paper. In the “selection of communication as a morbidity to measure” section, communication is referred to as a morbidity. Later in the results section, morbidities are identified as medical complications (e.g., acute neurological event, extracorporeal life support, feeding problems, etc.). This term should be utilized consistently throughout the document.
3. The sections “selection of communication as a morbidity to measure” and “selecting the questions about communication” are confusing and include information that does not add to the understanding of the methodology of the study. Please replace these sections with a short paragraph describing how and why communication was selected as the outcome as well as the operational definition for communication in this study (e.g., 6 Pickler questions).
4. At present, it is unclear why full questions were converted into binary questions. More information is needed on the question and response structure in the comparison Pickler study. If the response range was non-binary in either study, it is necessary to further clarify the method of data conversion.

5. The changes in wording (e.g., “always,” “definitely,” “completely”) used in the binary questions may have skewed results of the study, by only allowing for all or nothing response options. Without a wider range of response choices, the actual degree of communication is unclear (e.g., would expect differences for a family that was told consistent information 95% of the time vs. 30% of the time). Additionally, changes in wording also may have changed the meaning of certain questions (e.g., in Q3 no longer measuring confusion in binary question). If non-binary data is not available, this should be stated as a limitation, otherwise non-binary data is preferred for analysis.
6. Communication is defined differently in each of the 6 questions. For a more valid and reliable measure of communication and to prevent type 1 error, it may be useful to combine responses to these questions into one overall metric of communication prior to analyzing each individual item.
7. There was no information on parent involvement or availability. If a measurement of this construct is unavailable, it would be important to mention it as a limitation (e.g., if parents are less involved with care and/or not present during hospitalization, communication will likely be more limited).
8. Please clarify if Pickler sample was made up of controls and morbidity sample. If not separated, comparing groups separately to the larger sample will likely yield inaccurate results because populations should not be equivalent. Pickler sample should be separated out or impact sample should be grouped together prior to comparison.
9. Please include information about who filled out the survey (e.g., mothers, fathers, caregivers, etc.).

Results:

1. First paragraph of results section may be more appropriate in the methods section.
2. Please include statistics for non-significant results (e.g., explanatory factors associated with questions 5 and 6).
3. Consider running analyses to control for explanatory factors identified as relating to communication when comparing sample group to national sample. This will help clarify whether differences were due to communication variables or other factors (e.g., age, morbidity status)

Discussion:

1. The Discussion is short and about half of the content is reporting of results . The following comments include suggestions for how to expand upon the discussion.
2. As noted in the discussion, there are other factors that may influence various aspects of communication (e.g., age of child, morbidity status). Without more information about demographic features of participants across sites as well as additional analyses that control for these factors, claims about implications for communication are not necessarily justified.
3. Please include more information on study limitations (e.g., possible systematic differences between people in Pickler vs. impact study; binary response options; etc.) that could have influenced results.
4. To translate research to practice, future directions should include a measure of parental stress in order to make claims about clinical significance of findings. Just because differences were found does not necessarily indicate a need for change in practice. Please also indicate other future research directions.

	5 Please provide a possible explanation for non-significant results. References/Figures:  1. Good use of current references. 2. Please label figures. 3. Please include table with statistical results from full analysis. 4. Please include Q5 and Q6 in Figure 2 as well as data for all questions, even if not statistically significant.
--	--

REVIEWER	Reviewer name: Aditya Badheka, MD MS Institution and Country: University of Iowa Children's Hospital, Iowa City, IA, United States Competing interests: I have no competing interests.
REVIEW RETURNED	22-Dec-2018

GENERAL COMMENTS	Thank you Pagel et al for working on this important topic. The study is well done and showed interesting results. This may provide some insights into improving communication between care teams and families. Below are my comments/ suggestions:  1. Complex congenital heart disease care involves several specialty teams. However, ICU team communicate with the parents primarily. There is marked variability between centers in this study. It would be interesting to know the structure of the ICU team (especially rounding team) - the number of attending physicians, number of trainees (fellows/ residents/ Advanced providers), the number of social workers and presence of palliative care team. 2. Current literature demonstrated that ICU physicians are at very high risk of burn-out. The ratio between the number of ICU admissions and ICU physicians per center may shed a light on that issue. Thank you
---

VERSION 1 – AUTHOR RESPONSE

Reviewer 1

I confine my remarks to statistical aspects of this paper.

Since the patients were clustered within hospitals and since patients in a particular hospital are likely to share some similarities, the normal logistic regression model may not be appropriate. Table 2 shows large differences by site, so, the authors should use a nonlinear multi-level model.

I don't really like the binarization of questions. Surely most of these questions are not "always" vs. "not always" but vary along a continuum? Unless reason can be shown for binarization (e.g. that almost no patients gave any answer other than "always") this is a mistake that leads to loss of power and increased type I error.

Rather than test each question separately, it might make sense to combine them using factor analysis. This would greatly increase statistical power, not only because the combined measure would have less error but because the Bonferroni correction would be /2 rather than /12. In addition, this would make the response more-or-less continuous.

Taking the log of age does not really address the problem of nonlinearity. It would be better to use a spline of age.

Response: Following the editor's suggestion, we have rewritten the paper to be a descriptive analysis of the results only. We have also included all the responses for each question to give a fuller picture, which has required updating the figure presentation substantially.

Reviewer 2

This study aims to compare aspects of communication between families and medical teams for pediatric cardiac surgery patients to a national, all specialty survey. While communication is a very important aspect of care in pediatric cardiology practice, several concerns about methodology and study limitations limit the interpretability and utility of findings.

Abstract:

1. The abstract is likely to be confusing to a reader not already familiar with the overall study, impact substudy, and the various samples. Also, multiple paragraphs in one section of the abstract is not typical. Please consider editing for clarity.

Response: We have substantially rewritten the abstract which we hope addresses the reviewer's concerns.

2. The second point in "what this study adds" should be clarified and expanded upon. Confusion was not necessarily measured after the full question was changed into a binary question. This statement may be a misrepresentation of findings. It may be more accurate to say that 2/3 of parents reported that they were not always told consistent information.

Response: We have rewritten this point to read: "- It was more common for parents of children undergoing heart surgery to report being told different things by different people (53% said it happened sometimes and 10% said a lot) compared to the all-speciality survey (25% and 7% respectively)"

Introduction:

1. The literature review in the introduction provides a good rationale for importance of study. Aspects of communication identified in introduction link well with survey questions.

Response: We thank the reviewer for this positive comment

2. There is no justification provided for separating patients in the impact study by morbidity status. Please provide background literature to support this decision.

The larger study was explicitly designed to look for differences in impact between children who experienced a morbidity and those who did not. However, we agree that for this explorative secondary analysis on communication, it is not appropriate to prioritise disaggregating the sample by morbidity. We thus now present results for our entire sample compared to the national, all-specialty, survey first and then include presence of a morbidity as an additional characteristic in the secondary exploration.

3. Portions of the introduction would be better suited for the methods section (e.g., background on Pickler survey).

Response: We agree and have moved the details of the Pickler survey and the patient characteristics considered to the methods section.

4. Please include a statement about the knowledge gaps that this study intends to address. Why is this particular study needed? Please also include the research questions and any hypotheses in introduction.

Response: We have substantially rewritten this paper as a descriptive paper at the suggestion of the editor and we agree with the editor that this structure better fits the exploratory nature of this study. We have changed the last sentence of the introduction to read "Our aim was to explore how parents of children undergoing heart surgery perceived communication with their clinical team and the potential association of patient characteristics with reported quality of communication."

Methods:

1. The demographic information that was included in the paper is insufficient for fully understanding the results of this study. For example, who were the participants included in the Pickler survey (e.g., inclusion of "control" participants in Pickler sample? gender, age, length of hospital stay, etc. [include table of demographic information]; were patients with CHD included in Pickler sample?). Demographic information on participants in impact study are also critical (e.g., gender, number of morbidities, etc.) as well as a formal comparison of demographic characteristics of participants across 4 impact sites and Pickler study to assess for systematic differences that could have impacted the internal validity.

The Pickler survey included all children treated in England during a single month which we hope is clear from this sentence: "Questionnaires were sent to families of all children discharged from all hospitals in England during August 2014, with over 19,000 responses (response rate 27%) [17]."

Patient characteristics of the participants in our study are given in results (table 2). Characteristics of the non-responders have been added as a final column in Table 2.

Response: We present the comparison to the national survey for all patients in our survey (Table 3, Figure 1) – we no longer disaggregate by complication for this comparison. We consider complication status along with the other complications in tables 4 and 5 and figures 2-5.

We also hope that removing the emphasis on complication from this paper has made the analysis clearer to follow.

2. The use of the word "morbidities" is variable throughout the paper. In the "selection of communication as a morbidity to measure" section, communication is referred to as a morbidity. Later in the results section, morbidities are identified as medical complications (e.g., acute neurological event, extracorporeal life support, feeding problems, etc.). This term should be utilized consistently throughout the document.

Response: We now use the term "complications" throughout.

3. The sections "selection of communication as a morbidity to measure" and "selecting the questions about communication" are confusing and include information that does not add to the understanding of the methodology of the study. Please replace these sections with a short paragraph describing how and why communication was selected as the outcome as well as the operational definition for communication in this study (e.g., 6 Pickler questions).

These sections were originally contained within an appendix but were moved to the main paper at the request of the editors. We do believe that they help the reader understand the context of the communication survey and are reluctant to reduce this material.

We have changed the first subheading of the methods section to read “Larger study overview to select, define and measure complications following children’s heart surgery”. We hope that this helps to make the following two sections more obviously about the background to this study.

4. At present, it is unclear why full questions were converted into binary questions. More information is needed on the question and response structure in the comparison Pickler study. If the response range was non-binary in either study, it is necessary to further clarify the method of data conversion.

Response: We no longer convert the questions into binary and present the range of responses to each question.

5. The changes in wording (e.g., “always,” “definitely,” “completely”) used in the binary questions may have skewed results of the study, by only allowing for all or nothing response options. Without a wider range of response choices, the actual degree of communication is unclear (e.g., would expect differences for a family that was told consistent information 95% of the time vs. 30% of the time). Additionally, changes in wording also may have changed the meaning of certain questions (e.g., in Q3 no longer measuring confusion in binary question). If non-binary data is not available, this should be stated as a limitation, otherwise non-binary data is preferred for analysis.

Response: We have now presented the range of responses to each question.

6. Communication is defined differently in each of the 6 questions. For a more valid and reliable measure of communication and to prevent type 1 error, it may be useful to combine responses to these questions into one overall metric of communication prior to analyzing each individual item.

When selecting the questions with the Picker Institute we deliberately set out to capture a range of aspects of communication between families and clinical teams. We have made this clearer by changing the sentence in the methods section to read: “The definitions panel worked with the Picker Institute to identify 6 questions from their national survey to ask study parents about communication that captured a range of aspects of communication between families and clinical teams.”

Response: We prefer to keep the questions separate so we can explore these different aspects of communication. As we have now rewritten the paper as a descriptive study, the question of Type I error no longer applies.

7. There was no information on parent involvement or availability. If a measurement of this construct is unavailable, it would be important to mention it as a limitation (e.g., if parents are less involved with care and/or not present during hospitalization, communication will likely be more limited).

Response: We have changed the last section of the first paragraph of the discussion to read: “Possible explanations are that heart surgery is a particularly complex treatment involving a larger than average clinical care team and that a child’s course of recovery can be very variable. Additionally, children undergoing heart surgery have longer stays in hospital than the overall paediatric inpatient population and parents typically spend a lot of time by their child’s bedside, providing more opportunities for confusing communication to arise.”

8. Please clarify if Pickler sample was made up of controls and morbidity sample. If not separated, comparing groups separately to the larger sample will likely yield inaccurate results because populations should not be equivalent. Pickler sample should be separated out or impact sample should be grouped together prior to comparison.

Please see the response to point 1 above.

9. Please include information about who filled out the survey (e.g., mothers, fathers, caregivers, etc.).

Response: We do not know which parent or caregiver filled out the survey so unfortunately, we cannot include this information.

Results:

1. First paragraph of results section may be more appropriate in the methods section.

We agree and have moved this paragraph to the methods as suggested.

2. Please include statistics for non-significant results (e.g., explanatory factors associated with questions 5 and 6).

Response: We no longer perform any tests of significance, but we have now included a full set of responses for each characteristic for each question in tables 4 and 5.

3. Consider running analyses to control for explanatory factors identified as relating to communication when comparing sample group to national sample. This will help clarify whether differences were due to communication variables or other factors (e.g., age, morbidity status)

Response: We believe that now that this is a descriptive paper, this point no longer applies.

Discussion:

1. The Discussion is short and about half of the content is reporting of results. The following comments include suggestions for how to expand upon the discussion.

2. As noted in the discussion, there are other factors that may influence various aspects of communication (e.g., age of child, morbidity status). Without more information about demographic features of participants across sites as well as additional analyses that control for these factors, claims about implications for communication are not necessarily justified.

3. Please include more information on study limitations (e.g., possible systematic differences between people in Pickler vs. impact study; binary response options; etc.) that could have influenced results.

4. To translate research to practice, future directions should include a measure of parental stress in order to make claims about clinical significance of findings. Just because differences were found does not necessarily indicate a need for change in practice. Please also indicate other future research directions.

5. Please provide a possible explanation for non-significant results.

Response: We have now expanded the discussion, including a paragraph on limitations, which we hope addresses these points.

References/Figures:

1. Good use of current references.

2. Please label figures.
3. Please include table with statistical results from full analysis.
4. Please include Q5 and Q6 in Figure 2 as well as data for all questions, even if not statistically significant.

Response: We have redone all the figures and added more tables giving the full results.

Reviewer 3

Thank you Pagel et al for working on this important topic. The study is well done and showed interesting results. This may provide some insights into improving communication between care teams and families.

Response: We thank the reviewer for these positive comments.

Below are my comments/ suggestions:

1. Complex congenital heart disease care involves several specialty teams. However, ICU team communicate with the parents primarily. There is marked variability between centers in this study. It would be interesting to know the structure of the ICU team (especially rounding team) - the number of attending physicians, number of trainees (fellows/ residents/ Advanced providers), the number of social workers and presence of palliative care team.

Response: This is an interesting suggestion but unfortunately these data are not readily available and, if they were, might enable identification of the units. We have changed the penultimate sentence in the discussion to read "That improved communication is possible and worth further investigation of possible modifying factors is suggested by the consistently higher proportion of responses recorded at Site A across questions 1 to 4. For instance, the structure of ICU staffing, physical layout of a unit and the ratio of volume of admissions to staff may all affect parents' experience of communication with their child's clinical team."

2. Current literature demonstrated that ICU physicians are at very high risk of burn-out. The ratio between the number of ICU admissions and ICU physicians per center may shed a light on that issue.

We have added this is a possible factor in the discussion sentence quote above.

Comment: Thank you

VERSION 2 – REVIEW

REVIEWER	Reviewer name: Peter Flom Institution and Country: Peter Flom Consulting, USA Competing interests: None
REVIEW RETURNED	15-Feb-2019

GENERAL COMMENTS	I confine my remarks to statistical aspects of this paper. While other things could be done with this data, what was done was fine and I recommend publication.
---

REVIEWER	Reviewer name: Erica Sood, PhD Institution and Country: Nemours/Al duPont Hospital for Children Competing interests: None
REVIEW RETURNED	20-Feb-2019

GENERAL COMMENTS	Many aspects of this revised paper are much improved from the prior version. The authors were very responsive to the reviewers' comments. In particular, the change from binary categories to the full range of responses strengthens the paper substantially. While I understand the rationale for the change to a descriptive paper, I am not used to seeing descriptive papers that talk about one percentage being higher or lower than another without any test for statistical significance. I am concerned that differences in percentages that may not be statistically significantly different are described as such. For example, readers are likely to assume by reading the abstract that the differences reported reached statistical significance. The style of the methods section is also different from what I am used to. For example, the use of the first person ("we") single sentence paragraphs, and the long length of this section and of the subheadings within the section (for example, "Selection of communication as a complication to measure"). It seems to me that the methods section could be streamlined for readability. Lastly, the tables are larger than what I am used to seeing in a published manuscript.
--

VERSION 2 – AUTHOR RESPONSE

Reviewer 2

Many aspects of this revised paper are much improved from the prior version. The authors were very responsive to the reviewers' comments. In particular, the change from binary categories to the full range of responses strengthens the paper substantially.

Response: We thank the reviewer for their positive comments and agree that including the full range of responses has strengthened the paper.

Comment: While I understand the rationale for the change to a descriptive paper, I am not used to seeing descriptive papers that talk about one percentage being higher or lower than another without any test for statistical significance. I am concerned that differences in percentages that may not be statistically significantly different are described as such. For example, readers are likely to assume by reading the abstract that the differences reported reached statistical significance.

Response: We have added a sentence to the abstract to say "This was a descriptive study only" and amended the language in the results to make this clearer.

We have gone through the paper in detail and adjusted the language where necessary to avoid encouraging the reader to assume statistical significance. We have also added a sentence the methods stating "Note that as this is a descriptive study only, we do not suggest that our results have statistical significance."

We have also added this sentence to the results “We reiterate that in describing potentially interesting differences in the results, we are not asserting statistical significance.”

Comment: The style of the methods section is also different from what I am used to. For example, the use of the first person ("we") single sentence paragraphs, and the long length of this section and of the subheadings within the section (for example, "Selection of communication as a complication to measure"). It seems to me that the methods section could be streamlined for readability.

We have further streamlined the methods section and included a signposting paragraph at the start that explains what is contained in the methods and why:

“We have broken the methods section into subsections that cover:

- the larger study, including how the complications studied were selected for measurement and defined;
- how the communication survey was administered and to whom;
- how we analysed the results;
- ethical approvals and patient involvement.”

Lastly, the tables are larger than what I am used to seeing in a published manuscript.

We have amended Table 3 to be smaller and deleted tables 4 and 5 as suggested by the Editor in Chief.